# Identification of a quasi-liquid phase at solid–liquid interface

Xinxing Peng [1,2,5,6], Fu-Chun Zhu[1,6], You-Hong Jiang[1,6], Juan-Juan Sun[1], Liang-Ping Xiao[1], Shiyuan Zhou[1], Karen C. Bustillo [3], Long-Hui Lin[1], Jun Cheng [1], Jian-Feng Li [1], Hong-Gang Liao [1✉], Shi-Gang Sun[1] & Haimei Zheng [2,4✉]

An understanding of solid–liquid interfaces is of great importance for fundamental research as well as industrial applications. However, it has been very challenging to directly image solid–liquid interfaces with high resolution, thus their structure and properties are often unknown. Here, we report a quasi-liquid phase between metal (In, Sn) nanoparticle surfaces and an aqueous solution observed using liquid cell transmission electron microscopy. Our real-time high-resolution imaging reveals a thin layer of liquid-like materials at the interfaces with the frequent appearance of small In nanoclusters. Such a quasi-liquid phase serves as an intermediate for the mass transport from the metal nanoparticle to the liquid. Density functional theory-molecular dynamics simulations demonstrate that the positive charges of In ions greatly contribute to the stabilization of the quasi-liquid phase on the metal surface.

[1] State Key Laboratory for Physical Chemistry of Solid Surfaces, Collaborative Innovation Center of Chemistry for Energy Materials, College of Chemistry and Chemical Engineering, Xiamen University, Xiamen 361005, China. [2] Materials Science Division, Lawrence Berkeley National Laboratory, Berkeley, CA 94720, USA. [3] National Center for Electron Microscopy, Molecular Foundry, Lawrence Berkeley National Laboratory, Berkeley, CA 94720, USA. [4] Department of Material Science and Engineering, University of California, Berkeley, CA 94720, USA. [5] Present address: National Center for Electron Microscopy, Molecular Foundry, Lawrence Berkeley National Laboratory, Berkeley, CA 94720, USA. [6] These authors contributed equally: Xinxing Peng, Fu-Chun Zhu, You-Hong Jiang. ✉email: hgliao@xmu.edu.cn; hmzheng@lbl.gov

Solid–liquid interfaces are omnipresent in nature and are of key importance across the fields of science and technology[1,2]. The structure, composition, and distribution of chemical species, and their changes in response to external stimuli are closely related to the interfacial reactivity[3]. Solid–liquid interfaces play an important role in heterogeneous catalysis[4], energy conversion[5], corrosion protection[6], and sensors[7]. They can also greatly contribute to the functions of biological systems[8,9]. Many techniques have been developed to study solid–liquid interfaces[10–14], including infrared spectroscopy and Raman spectroscopy, and the research has enhanced our understanding of interfacial chemistry. However, for many systems, techniques that allow imaging of the interfaces with high spatial and temporal resolution are needed[12,15]. The development of in-situ aberration-corrected transmission electron microscopy (TEM) has enabled the study of dynamic processes at solid-vapor interfaces with atomic resolution[16]. Solid–liquid metal systems in vacuums have also been investigated[17–19]. However, in-situ high-resolution imaging of solid–liquid interfaces is challenging due to several issues[20]. For example, since liquids are not compatible with the high-vacuum environment in the electron microscope, liquid cells that can isolate the liquid sample from the environment have been developed. Unfortunately, the liquid cell samples are usually thick, offering a limited opportunity for high-resolution imaging of the solid–liquid interfaces when the thickness of interfaces is only a few nanometers. Therefore, the ability to achieve the atomic imaging of solid–liquid interfaces has been considered to be critical for surface science research.

Recent developments in liquid phase TEM that allow imaging through liquids using a liquid cell provide an opportunity to overcome the restrictions[20–24]. Liquid cell transmission electron microscopy (LCTEM) has already revolutionized the characterization of materials, enabling the direct visualization of the dynamic processes in a liquid phase with high spatial resolution[25,26]. In addition to the observation of materials morphology evolution and motion dynamics, LCTEM also enables the measurement of composition and structure changes.

Here, we present the real-time imaging and analysis of solid–liquid interfaces. We identify a quasi-liquid phase at the interface of a metal nanoparticle and aqueous solution. The quasi-liquid phase adsorbing onto the metal nanocrystal exhibits distinct fluidity while maintaining an average thickness of 2–3 nm. Mass transport through the quasi-liquid is also examined. We characterize the quasi-liquid phase using electron energy loss spectroscopy (EELS), energy-dispersive X-ray spectroscopy (EDS), and fluorescence spectroscopy. The interphase is found to have fluid-like elasticity and consists of a metal-ion rich amorphous phase with small In nanoclusters/nanocrystals. Moreover, density functional theory-molecular dynamics (DFT-MD) simulations indicate that the quasi-liquid phase is stabilized through the charge effects.

## Results and discussion

**Imaging and structure characterization of the quasi-liquid phase at the solid–liquid interface**. In-situ LCTEM experiments are conducted by sandwiching $InCl_3$ aqueous solution between two ultrathin silicon nitride chips (Supplementary Fig. 1). The $InCl_3$ aqueous solution is irradiated to initiate nucleation and growth of In nanocrystals, as high-energy electrons are known to generate solvated electrons to reduce $In^{3+}$ to In atoms[27]. We directly observe the formation of a quasi-liquid phase on the surface of the In nanocrystals in the aqueous solution. The interphase is relatively stable and exhibits fluidity (Supplementary Movie S1 and Supplementary Movie S2), which greatly differs from the behavior of a traditional core-shell structure in liquids and liquid-like metals[28–30]. A schematic diagram of the core-interphase structure formed inside the liquid cell is presented in Fig. 1a. Figure 1b displays a representative low-resolution TEM image of the In nanocrystals. The nanoscale interphase uniformly

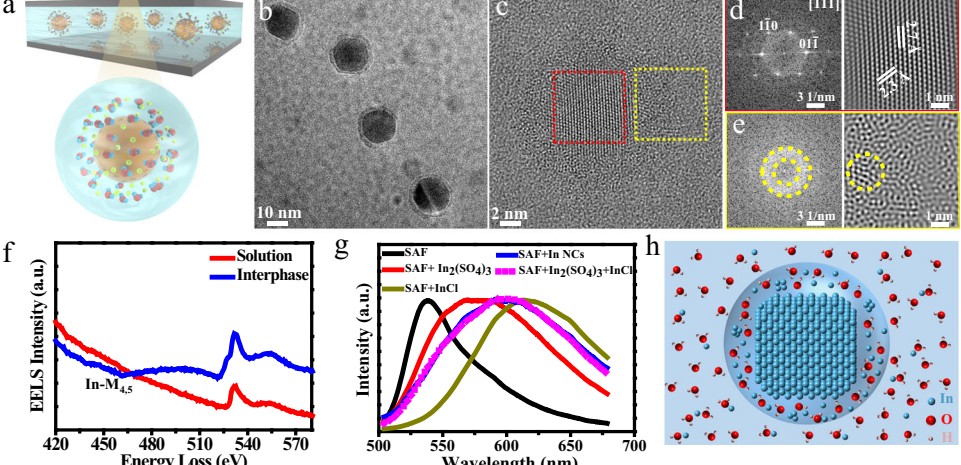

**Fig. 1 Imaging and structure characterization of the quasi-liquid phase at the solid–liquid interface in liquid phase TEM. a** Schematic illustration of In nanocrystals in aqueous solution. A quasi-liquid phase is observed on the surface of the In nanocrystals. The yellow, red and blue spheres represent In, O and H, respectively. The light blue, gray and yellow shading represent the liquid environment, liquid cell and electron beam, respectively. **b** Low- and **c** high-magnification TEM image of In nanocrystals in aqueous solution. **d** Fast Fourier Transformed (FFT) and Inverse Fast Fourier Transformed (IFFT) image from the core region enclosed in the red dashed square in **c**. **e** FFT and IFFT image from the solid–liquid interface region enclosed in the yellow square in **c**. The yellow ring area is the masked area for IFFT. TEM image analysis indicated that the core was crystalline, and the quasi-liquid phase was amorphous. **f** EELS spectra of the quasi-liquid phase (blue line) and surrounding solution (red line). In and O were both detected in the quasi-liquid phase. **g** Fluorescence spectra of SAF (black line) in the presence of In nanocrystals (blue line), $In_2(SO_4)_3$ (red line), InCl (golden line), and a mixed solution of $In_2(SO_4)_3$ and InCl (dashed pink line). The peak at 598 nm is attributed to the interaction between the interphase and SAF solution. **h** Model of In crystal with amorphous quasi-liquid phase in solution. The amorphous phase consists of In-ions, small In clusters, water and hydroxyl species.

distributes around the crystal core is imaged (Fig. 1b). From the high-resolution TEM image (Fig. 1c, Supplementary Fig. 2), an amorphous structure of the interphase is determined. The d-spacing 2.3 Å and 2.7 Å of the core measured in Fig. 1d are consistent with the (1-10) and (01-1) lattice planes of the In crystal structure viewed along the [111] direction, respectively. Interestingly, we find small In nanoclusters appearing within the amorphous quasi-liquid interphase from high-resolution TEM images, as shown by the yellow circle in Fig. 1e (see more details in Supplementary Movie S3 and Supplementary Fig. 3).

We further characterize the quasi-liquid phase on the surface of the In nanocrystals using EELS to determine the chemical composition. Figure 1f presents EELS spectra of the solution (red) and quasi-liquid phase (blue) collected from the corresponding positions, as shown in Supplementary Fig. 4. In the spectrum acquired from the area of the quasi-liquid phase, a broad peak starts at ~443 eV, which can be designated as In-$M_{4,5}$ edges. The strong O K-edge peak from the quasi-liquid phase implies that the quasi-liquid contains oxygen-containing compounds. It is noteworthy that the O K-edge of the interphase has a pre-edge at ~527 eV compared with the spectrum from the solution, which is different from the regular K-edge of oxygen in the water. The pre-edge of oxygen is attributed to the existence of hydroxyl species[31].

To further confirm that the In nanocrystals are surrounded by metal-ion rich interphase without electron-beam irradiation, we characterize In nanocrystals in salicyl fluorone (SAF) solution using fluorescence spectroscopy. SAF is used as a fluorescence dye, while $In^{3+}$ and $In^+$ are used as prototype samples. The In nanocrystals are synthesized by electro-deposition (Supplementary Fig. 5) and transferred into the SAF solution in an inert atmosphere to prevent their oxidation. Figure 1g presents the fluorescent spectra of the SAF solution in the presence of In nanocrystals, $In_2(SO_4)_3$, InCl, and mixed-solution ($In_2(SO_4)_3$ and InCl). A clear peak (598 nm) appears in the spectra between peaks for $In_2(SO_4)_3$ (575 nm) and InCl (615 nm), which matches the spectrum of the mixed solution. From the fluorescent spectra, we can infer that the quasi-liquid phase interacts with the SAF solution after the In nanocrystals are transferred. From the above analysis, we conclude that the core is made of In metal and the metal-rich quasi-liquid phase is water-mediated metastable In complexes (Fig. 1h).

Considering that all the in-situ experiments are conducted under beam irradiation, it is very important to determine the beam effect on the formation of interphase. Therefore, several control experiments are designed to determine that the formation of the quasi-liquid phase is caused by the interaction between In nanocrystals and solution and not by the electron-beam effect. We first confirm that the quasi-liquid phase can still be observed after the beam is shut off for 5 min. Moreover, it is important to note that the interphase is stable and maintained a certain thickness after leaving the liquid cell chip in the air for more than two weeks. In addition, a control experiment of In nanoparticle etching is performed to illustrate the influence of electron beam on the quasi-liquid phase formation. The quasi-liquid phase is well maintained at the initial 5 s and then disappears when the particle size becomes smaller (Supplementary Movie S4 and Supplementary Fig. 6). We consider that if the quasi-liquid phase arose from the reaction of metal surface with the radiolysis species from electron beam irradiation of water, the quasi-liquid phase shouldn't have disappeared. Additionally, we find that the thickness of the quasi-liquid layer is independent of the dose rate, which further confirms that the electron beam effect is not the dominant factor in the formation of the quasi-liquid layer (Fig. 2e and Supplementary Fig. 7). Therefore, we conclude that the formation of the quasi-liquid phase is not solely from the electron

beam effect, and other factors should be explored to understand its formation mechanisms.

We also study the interface between In@$In_2O_3$ nanocrystals and water to illustrate the formation conditions of quasi-liquid interphase. In@$In_2O_3$ nanocrystals are synthesized on the bottom window of the liquid cell by thermal evaporation (Supplementary Fig. 8). The bottom cell containing In nanocrystals is then combined with a top cell for in-situ observation. We observe that the behavior of In@$In_2O_3$ in water is completely different from that of the In nanocrystals in solution. The fluid-like interphase is not observed on the surface of the In@$In_2O_3$ nanocrystals. The In core can dissolve into the solution because of the oxidation species generated by electron-beam irradiation, eventually forming a hollow structure (Supplementary Fig. 9a). It is worth noting that some new In nanoparticles grow near the hollow particles. Interestingly, we could observe the quasi-liquid phase on the surface of these newly formed In nanocrystals (Supplementary Fig. 9b). These control experiments demonstrate that the quasi-liquid phase can only form on the surface of metal nanoparticles, and it does not form on metal oxide shells.

**Formation mechanism of the quasi-liquid phase.** To better understand the formation mechanisms of the quasi-liquid phase, we track the morphology of the interphase. The formation of the dynamic interphase during nanocrystal growth at the solid–liquid interface in an aqueous solution is investigated. Figure 2a shows the growth of In nanocrystals and the formation process of the quasi-liquid phase in the $InCl_3$ solution (Supplementary Movie S5). Three distinct stages of growth can be identified. During the first stage, the In nanocrystal grows rapidly through monomer attachment, and the projected area of In nanocrystal increases from 620 to 1074 $nm^2$ (Fig. 2a, c). In addition, the interphase is not detected during the growth of In nanocrystal in the first stage. Subsequently, the second stage of formation of the quasi-liquid phase occurs. After the projected area of the In nanocrystal increases to 1074 $nm^2$, the interphase appears in less than 0.2 s. Notably, the projected area of the core suddenly decreases at the moment when the interphase appears (indicated by the green arrow in Fig. 2c). Therefore, we conclude that the interphase formation is related to the etching of the core. As the concentration of surrounding radiation species and In-ions around In nanocrystal decreases and the interphase acts as a buffer layer, preventing direct contact between the solution and In metal core, the reactions between them slow down. Eventually, the interphase remained stable, and the projected area of the core does not show any obvious change in the final stage. We further study the quasi-liquid phase during the etching process of the core (Fig. 2b and Supplementary Movie S6). The core is oxidized into ionic species, which are incorporated into the quasi-liquid phase and diffuse out through the quasi-liquid interphase. Therefore, we conclude that the formation of the quasi-liquid phase is due to the etching of the core, and the stable thickness of the quasi-liquid phase is controlled by the mass-transfer process between In core and the solution. We find that the thickness of the interphase doesn't show obvious change as the diameter of the core decreases (Fig. 2d). To illustrate the relationship between the core and interphase, we collect the core diameter and interphase thickness for more than 80 nanocrystals. Statistical analysis reveals that the average thickness of the interphase is 2–3 nm. Interestingly, the interphase is not readily detected on the surface of In nanocrystals with diameters less than 10 nm. It is noted that In nanoparticles with the quasi-liquid phase used for statistical analysis are imaged at different electron beam

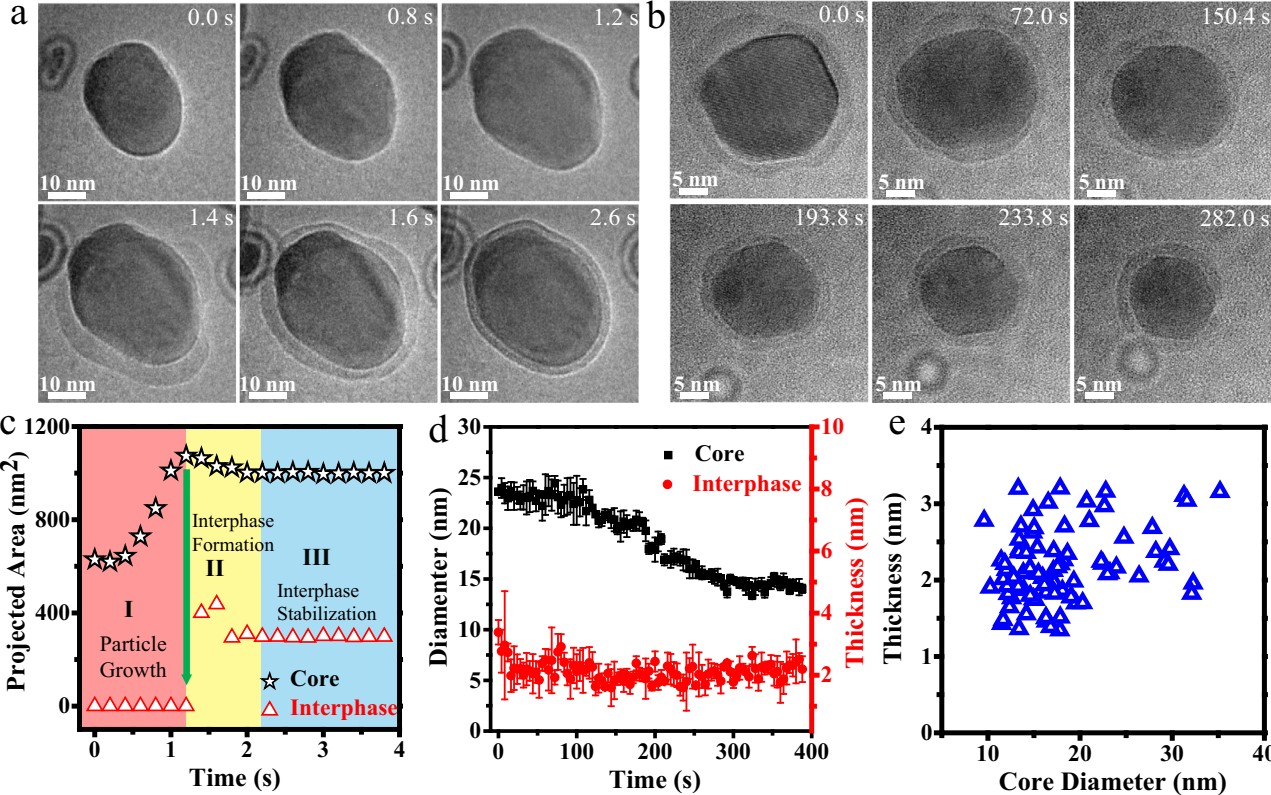

**Fig. 2 Formation and stabilization of quasi-liquid phase during In nanocrystal growth and dissolution. a** Sequential TEM images showing the growth of the In nanocrystal and formation of the quasi-liquid phase. Three distinct stages can be identified, namely the growth of the In nanocrystal (stage I), the appearance of the interphase (stage II) and the stabilization process of the interphase (stage III). **b** Sequential images showing no obvious change in the thickness of the quasi-liquid phase during the etching of the In core. **c** The projected area of the In core and interphase as a function of time, corresponding to **a**. The projected area of the core decreased at the moment when the interphase appeared (yellow area). **d** Diameter of the In core and thickness of the interphase as a function of time, corresponding to **b**. The error bars represent the difference between each set of data (three sets of data are measured at different positions of the same particle) and averaged diameter/thickness. **e** Statistical analysis of core diameter and the quasi-liquid phase thickness distribution for more than 80 nanocrystals. Source data are provided as a Source Data file.

dose levels (from 37.5 to 3750 $e^- \cdot Å^{-2} \cdot s^{-1}$), but the thickness of the interphase is independent of the dose rate.

In addition to the observation of the quasi-liquid phase on the surface of In nanocrystals, we also observe similar interphase on the surface of Sn nanocrystals when using $SnCl_4$ as the growth solution. HRTEM images reveal that the core is a tetragonal crystal, and the interphase is an amorphous structure (Supplementary Fig. 10 and 11). EDS results confirm that the quasi-liquid phase is mainly composed of tin, chlorine, and oxygen (Supplementary Fig. 10). In-situ observation of the generation of the Sn-related interphase also indicates that it is closely related to the etching of the core, as the area of the core decreases when the interphase appears (Supplementary Fig. 12). Our observation reveals that the core gradually transforms into the quasi-liquid phase, and the interphase dissolves into the solution to maintain a certain thickness (Supplementary Fig. 13 and Supplementary Movie 7). Moreover, we also observe that the interphase formed on the surface of the Sn nanocrystals in the $SnCl_4$ solution behaves like a fluid, differing from the behavior of $Sn@SnO_x$ in water. We do not observe any dynamic behavior of the amorphous $SnO_x$ in water, which also confirms that the interphase formed on the Sn nanocrystals in the $SnCl_4$ solution is not an amorphous oxide layer (Supplementary Fig. 14 and 15).

**Unique dynamic behavior and mass transfer process via the quasi-liquid phase.** This is the first time to our knowledge that such a noticeable quasi-liquid phase has been detected on the

surface of the metal nanocrystals in an aqueous solution. Further investigation is required to achieve a better understanding of the quasi-liquid phase. In particular, identification of the mass-transfer process in solution is of great importance as a clear demonstration of the connection between liquid and solid (i.e., the quasi-liquid phase). We observe many interesting phenomena resulting from the presence of the quasi-liquid phase. Metal clusters are observed to nucleate and grow directly from the quasi-liquid phase (Supplementary Movie S8). The free and fast phase change and mass transport inside one nanocluster or between different nanoclusters occurred within seconds. The growth of nanoclusters from the interphase is shown in Fig. 3a, c. A balance of growth and dissolution of the nanoclusters is observed in the interphase, and the diameter of the nanocluster changed periodically over time (Fig. 3b). After several cycles of growth and dissolution, nanocluster A finally disappears in the quasi-liquid phase. When nanocluster A disappears, nanocluster B grows rapidly nearby and escapes from the interphase, eventually dissolving in the solution. In addition to the nucleation and growth of the nanoclusters from the interphase, the coalescence process between the quasi-liquid phase and the surrounding nanocluster is also imaged (Fig. 3d, f). From Fig. 3e, it is apparent that the nanocluster size becomes larger all at once during insertion. With the increasing diameter of the small In nanocluster, the imaging contrast gradually decreases until it disappears in the interphase (Supplementary Movie S9). In the transition between the quasi-liquid phase and the In nanoclusters,

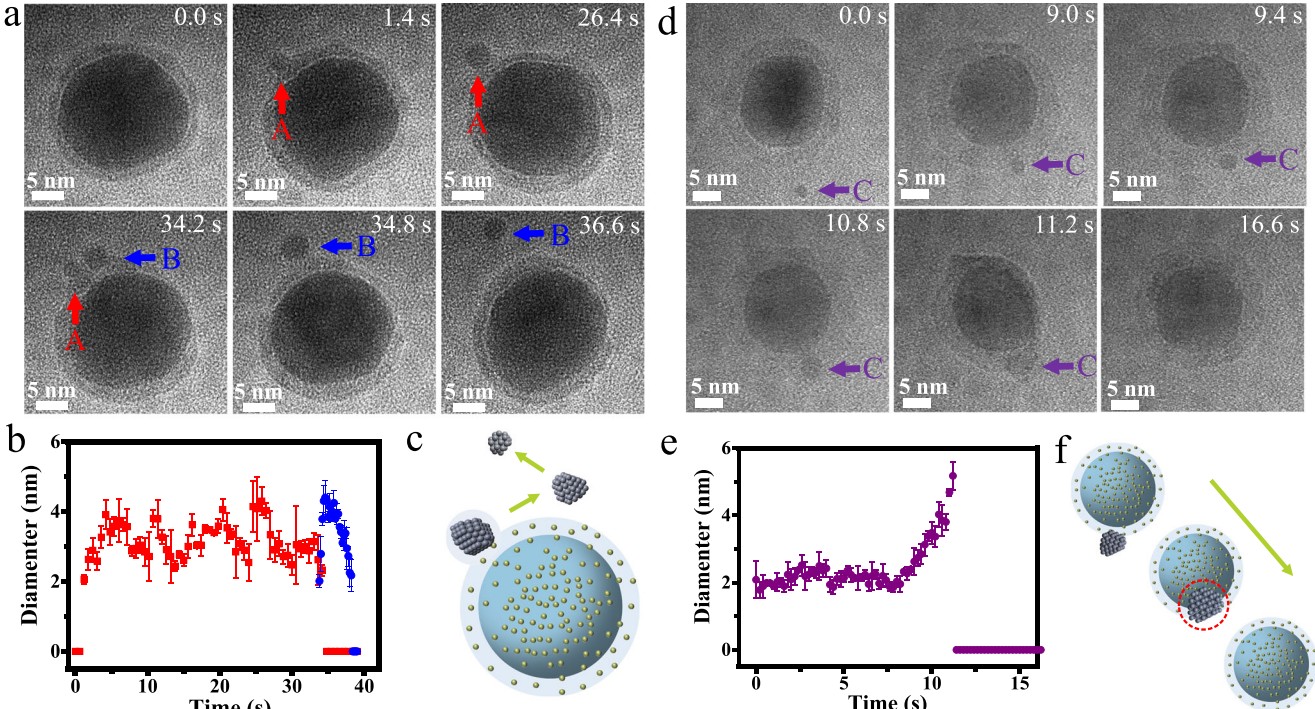

**Fig. 3 Dynamic behavior and mass transfer process via the quasi-liquid phase. a** Sequential in-situ TEM images showing the nucleation and dissolution of In nanoclusters in the quasi-liquid phase. **b** Time-dependent changes of the diameter of nanocluster A (red) and nanocluster B (blue) in **a**. **c** Schematic illustration of the nucleation of an In nanocluster from the interphase. **d** Sequential in-situ TEM images illustrating the coalescence process between the quasi-liquid phase and a small In nanocluster. **e** Time-dependent changes of the diameter of nanocluster C in **d**. **f** Schematic illustration of the coalescence process between the interphase and In nanocluster. The light blue shading and dark spheres in **c** and **f** represent interphase and In nanoparticles. The green arrows in **c** and **f** indicate the reaction direction. The error bars in **b** and **e** represent the difference between each set of data (three sets of data are measured at different positions of the same particle) and averaged diameter of small In cluster. Source data are provided as a Source Data file.

the main reaction is essentially based on the following equation: $2In + In^{3+} \leftrightarrow 3In^{+}$. When the equilibrium of the reaction is towards the left, we observe the precipitation of nanoclusters, and when the equilibrium of the reaction is towards the right, we observe the transition of the nanoclusters to the interfacial phase. The nucleation and dissolution of small nanoclusters indicate that the interphase acts as a medium for mass transfer between different species (metal core, surrounding particles) and solution.

**DFT-MD simulations on the stabilization of the quasi-liquid phase**. To understand the existence of a considerably stable quasi-liquid phase on metal surfaces, we perform the combined density functional theory (DFT) and molecular dynamics (MD) simulation. We hypothesize that the positive charges from In ions in the quasi-liquid phase might have contributed to the stability of the quasi-liquid phase. We estimate the proportional relation of In species and $H_2O$ molecules is approximately 1:2 in the interphase based on the thickness contrast of the HAADF-STEM images[24] (Supplementary Fig. 16). To verify the hypothesis, we establish models by considering the indium species under different states, In atoms versus In ions.

In the first scenario, we build a model containing 40 In atoms and 80 $H_2O$ molecules in a cubic box with a length of 15.31 Å. The initial structure is shown in Supplementary Fig. 17a, and the intermediate structures at 1, 4, 7, and 10 ps are shown in Fig. 4a (more intermediate structures are presented in Supplementary Fig. 18a). Our simulation shows that most In atoms rapidly gather together to form clusters in a process similar to spontaneous nucleation (Fig. 4a and Supplementary Movie S10). Only a small amount of In atoms disperse in the solution

and interact with $H_2O$ molecules. The charge of the In atom is mostly distributed between –0.8 and 0.2 $e^{-}$ (Fig. 4b and Supplementary Fig. 19a). Our simulations also indicate that the charge of the In atoms changes as their coordination number changes. The charge of the In atoms (e.g., $In_{254}$) becomes positive as the coordination number decreases; therefore, the scattered In atoms (e.g., $In_{250}$) are positively charged, whereas the aggregated In atoms (e.g., $In_{262}$) are negatively charged. The total charge on the 40 In atoms is approximately zero.

In the second scenario, we consider quasi-liquid phase contains In ions instead of In atoms. This can be achieved by removing 40 H from the In/$H_2O$ system in our simulation. The remaining OH tend to get electrons from In atoms to form stable hydroxide ions, thus it induces a positive charge on In (Supplementary Fig. 17b and Supplementary Movie S11). The intermediate structures of the DFT-MD simulations under this condition at 1, 4, 7, and 10 ps are illustrated in Fig. 4c (more intermediate structures are shown in Supplementary Fig. 18b). Compared with the dynamic behavior observed in the first scenarios with In atoms, the phenomenon in this case with In ions differs. There is no aggregation trend for most of the In ions during the entire process (Supplementary Movie S11). Some In can form small clusters in the local area owing to thermodynamic fluctuations. Considering that there are only 40 In ions in this model, we consider that these clusters can grow if there are more In ions. These randomly formed clusters are consistent with the observation of small In nanocrystals in the quasi-liquid phase in our liquid cell TEM experiments (Fig. 3a). Our simulation further shows that the charges of the In species are mostly distributed in the range of 0.2–0.6 $e^{-}$ (Fig. 4d and Supplementary Fig. 19b).

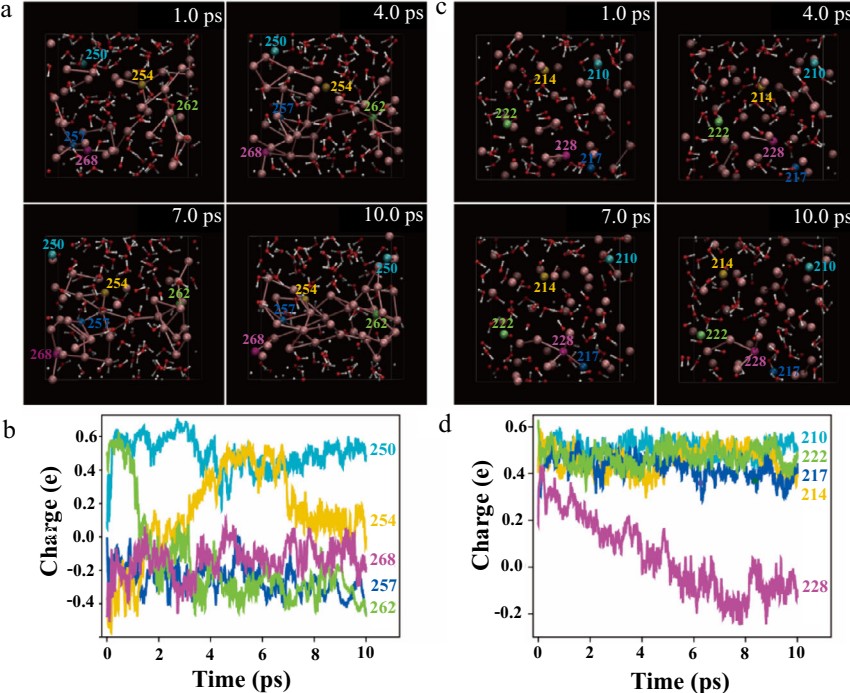

**Fig. 4 Density functional theory-molecular dynamics simulations on the stabilization of the interphase. a** Snapshots of the trajectory (1, 4, 7, 10 ps) of the dynamic structure under the electrically neutral condition. The electrically neutral model contained 40 In atoms (labeled from 240 to 279) and 80 $H_2O$ molecules in a cubic box with a length of 15.31 Å. **b** Charge of selected In atoms as a function of time under the electrically neutral condition. These atoms are labeled in a. **c** Snapshots of the trajectory of the dynamic structure (1, 4, 7, 10 ps) under positively charged condition. The In atoms are labeled from 200 to 239. **d** Charge of the selected In atoms as a function of time under positively charged condition. These particles are labeled in **c**. Atom color code: red, O; white, H; Other colors, In.

By comparing the 40 In atoms in the final structure in the above two scenarios, we find that 50% In atoms are aggregated when the quasi-liquid phase contains In atoms, while only 2.5% of In species are aggregated when the In species in the quasi-liquid phase are In ions. This simulation results are not surprising since the positive charges make it difficult for In species to gather together, thereby making the quasi-liquid phase relatively stable.

In summary, we have discovered a distinguishable quasi-liquid phase on the surface of metal nanocrystals in the aqueous solution. The quasi-liquid phase dynamically changes the configuration while maintaining an average thickness of 2–3 nm. Such a quasi-liquid phase is composed of a high concentration of metal ions and nanoclusters, thus different from the traditional understanding of a solid or liquid phase. Our DFT-MD simulation suggests that the positive charges of In ions within the quasi-liquid phase are responsible for stabilizing the interphase. Our study provides critical insights on solid–liquid interfaces, thus important to the synthesis and applications of metal nanoparticles related to interface reactions, such as catalysis and corrosion.

## Methods

**Materials**. All the chemicals including indium chloride (99.999%, Sigma-Aldrich), tin(IV) chloride pentahydrate (98%, Sigma-Aldrich), and In and Sn beads (99.99%, Kurt J. Lesker) were used as received. Ultrathin carbon film (10 nm, 400 mesh) supported gold grids were purchased from Electron Microscopy Sciences.

**Liquid cell fabrication and the growth-solution loading for TEM**. The liquid cell was fabricated using a method similar to that reported in our previous study[32]. First, a very thin silicon nitride membrane (10 nm thick) was deposited on a thin silicon wafer (4 inches, p-doped) purchased from Virginia Semiconductor (Fredericksburg, VA). The ultrathin silicon nitride membrane helped to improve the resolution. For the top chip, lithographic patterning and hot KOH solution etching of silicon were conducted. For the bottom chip, in addition to the lithographic

patterning and KOH etching, indium film (~100 nm) was deposited on the silicon nitride membrane by thermal evaporation, which could also serve as a bonding agent and spacer. We assembled the bottom chip and top chip under an optical microscope. Then, a small amount (~100 nL) of the solution was injected into one of the cavities in a liquid cell. The solution was sucked into the cell by capillary force and formed a thin liquid layer (~100 nm) sandwiched between two inorganic membranes. The liquid cell was subsequently sealed with epoxy. After 2 h, the cell was used as a standard TEM sample for imaging. For the growth of In and Sn nanocrystals with quasi-liquid phase, 20 mg/ml $InCl_3$ and $SnCl_4 \cdot 5H_2O$ aqueous solution were used, respectively. We also use a carbon-film liquid cell to image the growth of In and Sn nanocrystals.

**In@In₂O₃ and Sn@SnOₓ nanocrystals in H₂O using a carbon-film liquid cell**. The carbon-film liquid cell was prepared using the procedure reported in a previous study[33]. In nanocrystals with oxide shells were synthesized on the carbon film of TEM grids by thermal deposition. Thermal evaporation of In beads was executed in a high-vacuum resistance evaporation coating machine (ZHD300, Beijing Technol Science Co., Ltd.). The thermal evaporation was conducted under vacuum of $5.5 \times 10^{-5}$ Pa. The evaporation rate was 0.15 nm/s, and the deposition thickness was set to 10 nm. Because In nanocrystals are rapidly oxidized in the air, we will eventually get particles with oxide shells ($In@In_2O_3$). The synthesis of $Sn@SnO_x$ was performed using the same process as that for $In@In_2O_3$. The liquid loading and cell assembly were completed in air. We first drop a small amount of liquid on the top of the grid with nanoparticles, then covered it with another grid to form a sandwich structure.

**Electro-deposition of In nanocrystals**. The electro-deposition of the In nanocrystals was performed using a three-electrode cell at room temperature. The working electrode was a glassy carbon electrode (GCE). The electrolyte consisted of 0.5 M sulfuric acid and 0.1 M $InCl_3$. Before the electro-deposition process, the electrolyte was deoxidized by blowing nitrogen for more than 30 min. The three-electrode cell was placed into an airtight box equipped with a wire conduit and a gas-guide tube to create an air-free environment. The In nanocrystals were synthesized by applying −1.2 V (vs. SCE) for 1200 s. After the electro-deposition, the GCE was washed and transferred under $N_2$ condition.

**TEM characterization**. In-situ TEM was performed using a high-resolution transmission electron microscope (TECNAI F-20 and Talos) at Xiamen University. Dry samples were characterized using high-resolution TEM, high-angle annular

dark-field scanning transmission electron microscopy and energy-dispersive X-ray spectroscopy (Themis) at the Molecular Foundry, Lawrence Berkeley National Laboratory.

**Fluorescence spectrum characterization**. The fluorescence spectrum characterization was performed using a Raman-11 instrument (Nanophoton, Japan), equipped with 488 nm, 532 nm and 785 nm lasers. The excitation wavelength was 488 nm (the power ND filter was 50/255, and the exposure time was 1 s/shot). A 50×, NA 0.45 objective was used to focus on the surface for the acquisition of all the spectra.

To collect the fluorescence signals of $In^{3+}$ dyed by salicyl fluorone (SAF), $In^+$ dyed by SAF, and the standard SAF sample, we prepared SAF-$In^{3+}$ (0.28 and 0.5 mg/ml, respectively), SAF-$In^+$ (0.28 and 0.5 mg/ml, respectively), and the standard SAF solution (0.28 mg/ml). We spread a drop of these samples on the surface of every bare GCE. Before the liquid dries, the fluorescence spectra were immediately measured.

To collect the fluorescence signals of the experimental sample (In nanocrystals synthesized by electro-deposition), we spread a drop of the standard SAF solution (0.28 mg/ml) on the surface of the GCE with In nanocrystals. In the same way, before the drop dried out, fluorescence spectra were immediately measured. In addition, we further tested the blank control sample of a bare GCE that had been infiltrated into the electrolyte and undergone the same washing process as the electro-deposited In sample. The fluorescence spectrum was identical to that of the standard SAF sample.

**Computational method**
*Model building*. To study the effect of charge on In enrichment, we constructed a neutral system and a charged system. For the neutral system, 40 In atoms (labeled from $In_{240}$ to $In_{279}$ to track the charge change for each In atom) were uniformly dispersed in 80 water molecules. For the charged system containing In-ions, we removed 40 H in the In/$H_2O$ neutral system to induce a positive charge on In (the interphase contained In ions with different charges). The 40 In atoms are labeled from $In_{200}$ to $In_{239}$. Both the neutral and charged systems were put in cubic supercells with the full 3D periodic boundary conditions, and the box size was $15.31 \times 15.31 \times 15.31$ Å$^3$. The initial structure of the neutral system consists of 40 In atoms and 80 $H_2O$ molecules, and the initial structure of the charged system consist of 40 In atoms, 40 $H_2O$ and 40 OH molecules. Both the neutral system and the charged system are electrically neutral because hydroxides are formed, thereby balancing the charges.

*Computational setup*. To simulate the effect of charge on In enrichment, DFT-based ab initio molecular dynamics simulations were performed using the freely available program package CP2K/Quickstep[34]. The AIMD simulation temperature was controlled at 330 K in the canonical ensemble condition (NVT). We employed the Nosé–Hoover thermostat with a time step of 0.5 fs. For the DFT calculations, the Perdew–Burke–Ernzerhof functional[35] with Grimme's dispersion correction[36] was used. The metal core electrons were represented by analytical Goedecker–Teter–Hutter pseudopotentials[37,38]. The basis sets for the valence electrons ($4d^{10}5s^25p^1$ for In, $2s^22p^4$ for O, $1s^1$ for H) were short-ranged (less diffuse) double-ζ basis functions with one set of polarization functions[39].

## Data availability
All TEM data supporting the findings of this study are contained in the paper and its Supplementary Information files. All other relevant data are available from the corresponding author (H. Zheng) on request. Source data are provided with this paper.

## Code availability
All codes supporting the findings of this study are available from the corresponding author upon request.

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

## Acknowledgements

The work at Lawrence Berkeley National Lab was supported by the U.S. Department of Energy (DOE), Office of Science, Office of Basic Energy Sciences (BES), Materials Sciences and Engineering Division under Contract No. DE-AC02-05-CH11231 within the KC22ZH program. The work at Xiamen University was supported by the National Key Research and Development Program of China (2016YFB0100202) and the National Natural Science Foundation of China (Grants 21673198, 21673194, 21621091, 21401049 and 51272071). We used the electron microscopy facility at the Molecular Foundry, which was supported by the Office of Science, Office of Basic Energy Sciences, of the U.S. Department of Energy under Contract No. DE-AC02-05CH11231. We are very grateful to Yan-Xia Jiang, Yan-Yun Zhai, Jia-Yao Lin, Zai-Chun Zhu and Jun-Yu Zhang for their help in experimental design, figure modification and manuscript editing.

## Author contributions

X.P. and F.Z. performed in-situ TEM imaging. F.Z. captured the growth of the quasi-liquid phase in solution. Y. J., L.X., S. Z. and X.P. conducted ex situ TEM imaging, EELS and EDS. L.L. and J.L. did the fluorescence spectrum characterization. J.S. and J.C. performed the simulations. X.P. and F.Z. performed the data analysis in consultation with H.L. All authors discussed the results. X.P. and F.Z. wrote the manuscript with inputs from all the other authors. H.L. and H.Z. conceived and supervised the research.

## Competing interests

The authors declare no competing interests.
