## [Peer Review File · Nature Communications]

REVIEWER COMMENTS

Reviewer #1 (Remarks to the Author):

This paper reports an interesting and novel observation in LCTEM, of a quasi-liquid layer on metal nanoparticles (In, Sn) in liquid. They mainly take advantage of their liquid cell design with thin spacing (120 nm) and thin window (10 nm) to achieve high-resolution imaging in liquid, and be able to identify the 2-3 nm layer on the particles. The experiments are very thoroughly done, with careful characterization of the liquid phase composition, ebeam effects, control experiments with non oxide-coated nanoparticles, and careful DFT-MD simulation. I would recommend its publication after revisions to address the following comments.

Overall, the point that this layer can form without e-beam is not clearly illustrated. Compared to other LCTEM studies with metal particles, this work uses a rather clean liquid solution with only InCl_3 and no additional ligands. Thus one reasonable origin of this quasi liquid layer can be from the reaction of the clean metal surface with radiolysis species. More detailed comments are listed below:

1. The formation process of this quasi-liquid layer in Movie S3 is very impressive, but the 'bare' particles are kind of suggesting this quasi-liquid layer can be absent, even for diameter > 10 nm cases. Could the author clarify that.
2. For discussion on electron-beam effect: (1) In fluorescent spectra sample preparation, it seems hard to rule out the possibility that In ions are attaching to In NC surface and bring to SAF solution. (2) The author mentioned, "quasi-liquid phase can be observed after beam off 5 min". But is 5 min long enough to dissipate the radiation effect while the liquid is sealed? Or the quasi-liquid phase is formed again very fast? More supporting data might be needed here.
3. For further experiments of In NP etching, small particle nucleation, growth, and coalescence, if the thickness of the layer is independent of dose rate, it might also be a good supportive data.
4. In the calculation part, the assumption of $\text{In:H}_2\text{O}=1:2$ is very rough, having too little discussion and no reference. Could the authors provide more information?

Reviewer #2 (Remarks to the Author):

The authors use high-resolution liquid-cell TEM to capture the behaviour of the few nanometre thick solid-liquid layer around In nanoparticles in aqueous solution. The quality of the imaging data is impressive, with lattice resolution frequently observed, allowing for clear delineation between the crystalline nanoparticles and the amorphous interface layer. They capture the formation of the layer during nanoparticle seeding and growth, and mass transfer from the layer. These are all interesting and well presented results, with insightful modelling and analysis supporting their findings.

Some comments and questions:

- Is there a particular reason that In was chosen over another system? Other seed solutions could also reduce out metal nanoparticles following electron beam irradiation.
- Line 78,79, page 4 suggests that the contrast features indicated in Fig 1e are small In clusters, per the reconstructed FFT. Currently this seems an overly confident analysis, at least based exclusively on the single reconstructed image. Random darker contrast features can be seen along the right side of the IFFT as well, for instance. Could these not be a thickness effect? Would we expect to see signals from such small clusters of In over the contributions from the thickness of the water layer and windows? Their analysis here is supported somewhat by their later spectroscopy, however the current ordering of results suggests that the image analysis on its own is sufficient at identifying the interphase as containing In clusters.
- For Fig 1f, some more detail on the EELS acquisition would be useful. Is there an image of the region from where it was acquired? How was motion of the nanoparticle mitigated against during spectra acquisition?
- There is an interesting discussion on page 5 to 6 regarding the role of the electron beam on the interphase formation, and the absence of the interphase on ex-situ prepared oxide core-shell nanoparticles. Would it be possible to introduce metallic In nanoparticles ex-situ, similarly to how they have already done with their oxides, rather than forming them in-situ by the reduction of the seed solution? This would allow the authors to distinguish between the interphase forming due to some interesting seed growth behaviour in the solution, versus it being a consequence of the surface being metal or oxide.

Reviewer #3 (Remarks to the Author):

I should note at the outset that I have little expertise in the experimental techniques employed in this research, but, that said, I find it impressive that atomistic details of a solid-liquid interface can be seen in real time.

Possibly because of the word limit and because the authors naturally want to report all the necessary checks they made, it was not always easy to get the main points of the story. Just getting a clear account would be my main recommendation. That said, here are some queries which I would encourage the authors to consider.

1) The three stages mentioned are a) the formation of an In cluster, b) the formation of quasi-liquid phase and the etching away of the core, and c) a steady state in which both the interfacial properties and the projected interfacial area of the core remained roughly constant. I am still not clear about this final stage, however. Earlier the authors stated that the interface retained its thickness up to a time of two weeks, so at this point I assumed the system was at equilibrium. Figure 2d shows the diameter of the core continually decreasing up to 400 s, so clearly the system is not in equilibrium up to that time. Can the authors clarify what the equilibrium state is? Or is equilibrium a state in which all the clusters have been etched away?

2) I am also not clear what chemical reaction is thought to be going on. The authors argue that hydroxide is present in the interfacial region. Is the In forming some kind of cation in order to go into solution? Some clarity here would certainly help me.

3) The authors discuss the liquid like material showing dynamical changes between amorphous and nano-crystalline. I did stare at the videos but found it hard to see where/when this was happening. The figures in the manuscript did not make this obvious, at least not to me. This is an intriguing observation but does need some backing up with evidence. It would also be nice to know what species were involved, something on the timescales of the process, etc.

4) The computer simulation was interesting but some more details would be helpful. Firstly how did the authors verify the methodology (e.g. how good is the functional, how confident are they that they are working at a reasonable pressure, etc)? Secondly the simulation videos had lines appearing and disappearing and some sort of colour scheme. these should be explained.

5) More explanation should be given to the charged simulation. The starting configuration would appear to be In and OH⁺ ions, from the text, though I am not sure that is what was in the video. It does seem odd to have OH⁺, though. But even so, I presume the system eventually settles down but in a way that does not resemble the experimental system which is charge neutral, containing chloride ions. On the face of it, this is a very artificial set-up and I am not clear what conclusions can be drawn from it.

6) Just a minor query on the uncharged simulation, is the total charge on the In atoms approximately zero or is there net charge transfer to the water? Do you get hydroxide forming, for example? And again, how useful is this simulation, given the absence of chloride?

I apologise if the answers to many of these questions are obvious, but I think the authors need to make everything a lot clearer before publication, even though I think the experimental observations are, to my mind, impressive.

Reviewer #1

This paper reports an interesting and novel observation in LCTEM, of a quasi-liquid layer on metal nanoparticles (In, Sn) in liquid. They mainly take advantage of their liquid cell design with thin spacing (120 nm) and thin window (10 nm) to achieve high-resolution imaging in liquid and be able to identify the 2-3 nm layer on the particles. The experiments are very thoroughly done, with careful characterization of the liquid phase composition, ebeam effects, control experiments with non oxide-coated nanoparticles, and careful DFT-MD simulation. I would recommend its publication after revisions to address the following comments.

[Re] We thank the referee for his/her positive comments and valuable suggestions. We provide point-to-point responses as follows.

Overall, the point that this layer can form without e-beam is not clearly illustrated. Compared to other LCTEM studies with metal particles, this work uses a rather clean liquid solution with only InCl_3 and no additional ligands. Thus one reasonable origin of this quasi liquid layer can be from the reaction of the clean metal surface with radiolysis species.

[Re] Thanks for the comments. From Movie S5, we can observe that the projected area of the metal-core suddenly decreases at the moment when the quasi-liquid phase appears (Fig. 2c). This refers that the interphase formation is related to the etching of the core. The oxidation species formed under electron beam irradiation contributed to the surface In atoms becoming In-ions species, followed by adsorbing on the metal surface to form the quasi-liquid phase. However, the In-ions adsorption can still occur without electron beam or at low electron beam current intensity. Interfacial tension is used to describe the interaction between the metal core and adsorption layer in solution. The main forces involved in the interfacial tension are the adhesive forces. The adhesive force could be either dispersive adhesion (physisorption) or chemical adhesion (chemisorption). In our system, the positively charged In-ions are adsorbed on the surface of the In metal nanoparticle to reach an equilibrium stage with a nanoscale quasi-liquid phase. Therefore, we consider that radiolysis species could help the formation of the quasi-liquid phase, but it is not the direct cause. Following the referee's comment, we have added more discussions on the electron beam effect on the quasi-liquid phase formation (highlighted in pages 5 and 6)

More detailed comments are listed below:

1. The formation process of this quasi-liquid layer in Movie S3 is very impressive, but the 'bare' particles are kind of suggesting this quasi-liquid layer can be absent, even for diameter > 10 nm

cases. Could the author clarify that?

[Re] During the growth process of In crystals, bare nanocrystals are also observed even the diameter of the particles is larger than 10 nm. In this process, the In nanocrystals are in a non-equilibrium state. Large amounts of In atoms form and diffuse to the surface of In nanocrystals, which prevents the adsorption of In-ions to form the quasi-liquid phase. Besides, the concentration of In-ions will decrease sharply around the In nanocrystal because of the reduction reaction from In-ions to In atoms. Therefore, some particles without quasi-liquid interphase can be found even when the particle size is larger than 10 nm during the growth process.

2. For discussion on electron-beam effect: (1) In fluorescent spectra sample preparation, it seems hard to rule out the possibility that In ions are attaching to In NC surface and bring to SAF solution.

[Re] Thanks for the comment. In fact, the formation of the quasi-liquid phase is related to the adsorption of In-ions species on the surface of In nanocrystals. Therefore, we agree that “it’s hard to rule out the possibility that In-ions are attaching to the In nanocrystal surface”. However, we do not think a large number of free In-ions will be brought to the SAF solution after the synthesized nanoparticles on the electrode are washed with deionized water several times in our study.

3. The author mentioned, "quasi-liquid phase can be observed after beam off 5 min". But is 5 min long enough to dissipate the radiation effect while the liquid is sealed? Or the quasi-liquid phase is formed again very fast? More supporting data might be needed here.

For further experiments of In NP etching, small particle nucleation, growth, and coalescence, if the thickness of the layer is independent of dose rate, it might also be a good supportive data.

[Re] Due to the radiolysis of water under the electron beam, radical and molecular species including e_h^- , hydrogen radicals ($H\cdot$), hydroxyl radicals ($HO\cdot$) and H_2 can be generated^{1,2}. The reaction specie, such as e_h^- , is a free-electron surrounded by water molecules. These reactive species usually annihilate very quickly³. For example, the solvated electrons⁴ and free radicals⁵ have a lifetime at the scale of microseconds or even shorter in the liquid environment. So, five mins would be enough to dissipate the radiation species unless the bulk solution chemistry has been changed due to long-time irradiation⁶. We also did a control experiment by leaving the liquid cell containing In metal nanoparticles with quasi-liquid interphase for more than two weeks, and we found that the interphase was stable and maintained a certain thickness.

From our in-situ movies, the formation of the quasi-liquid phase happens within seconds. The time resolution of our camera is milliseconds. If the quasi-liquid phase is formed again, we should be

able to capture this process.

To further confirm the thickness of the layer is independent of the dose rate and is determined by the properties of the particles themselves, we have done additional experiments. As shown in Supplementary Fig. 7, no obvious thickness change of the quasi-liquid layer can be found when different electron dose rates are used. In fact, it has already been demonstrated in many nanoparticles for statistical analysis of the quasi-liquid layer thickness (Fig. 2e), the quasi-liquid interphase has a narrow thickness distribution of 1–4 nm (mainly distributed in the range of 2–3 nm), regardless of the electron beam dose.

Supplementary Fig. 7 TEM images of the same core-interphase particle at different magnification. The images from low to high magnification correspond to the dose rate change from low to high dose accordingly. The dose rates used for these four TEM images are 73.3, 161, 285 and $559 \text{ e}^- \cdot \text{\AA}^{-2} \cdot \text{s}^{-1}$, respectively.

The etching of In nanocrystal with quasi-liquid phase was performed to further support our claim. The In nanocrystal etching process shows that the quasi-liquid phase is well maintained at the initial 5 s, and the quasi-liquid phase disappears when the particle becomes smaller (Movie S4 and Supplementary Fig. 6). If the quasi-liquid phase is only formed because of the irradiation beam

effect, the quasi-liquid phase would not disappear when the nanoparticle is irradiated at a high dose rate of $994 \text{ e}^- \cdot \text{\AA}^{-2} \cdot \text{s}^{-1}$. Therefore, we conclude that the formation of the quasi-liquid layer has a relationship with the properties of nanocrystals themselves. The instability of quasi-liquid phase on small In nanoparticles worth future study.

Supplementary Fig. 6 In-situ observation of the etching of In nanocrystal with quasi-liquid phase. The quasi-liquid phase disappears when the particle size of In becomes smaller (e.g., the image at 16.6 s)

4. In the calculation part, the assumption of In: H₂O=1:2 is very rough, having too little discussion and no reference. Could the authors provide more information?

[Re] Thanks for raising this important question. The atomic concentration of In in the liquid cell TEM can be roughly estimated from the thickness contrast of the HAADF-STEM images⁷ as shown in Supplementary Fig. 16. We use a concentric ball model as shown in Supplementary Fig. 16a (the inner yellow ball represents the inner In metal core and the outer blue ball represents the outside interphase). So, we collect four directions (top, bottom, left and right) intensity line of the tangent position of two balls (this means the double thickness interphase mass contrast plus the background contrast). And the transverse and lengthways intensity line across the center of the two balls (this means the inner In core mass contrast plus the double interphase mass contrast, and plus the background mass contrast). Based on the STEM image contrast, we can calculate the atomic concentration of In element in the interphase is about 27.72% to 43.35% relative to In metal. The averaged In atomic concentration of the quasi-liquid phase is 33%, which is around three times

higher than the contrast of bulk solution. Since we only consider the existence of In and H₂O in the theoretical model, the assumption of In: H₂O is close to 1:2.

Supplementary Fig. 16 The calculation of atomic concentration of In through the model specification. **a** The model specification of In core and interphase. **b** HAADF-STEM image of the In nanocrystal in the liquid cell. **c** The mass contrast intensity curves of different directions are marked in a.

Reviewer #2

The authors use high-resolution liquid-cell TEM to capture the behavior of the few nanometers thick solid-liquid layer around In nanoparticles in aqueous solution. The quality of the imaging data is impressive, with lattice resolution frequently observed, allowing for clear delineation between the crystalline nanoparticles and the amorphous interface layer. They capture the formation of the layer during nanoparticle seeding and growth, and mass transfer from the layer. These are all interesting and well-presented results, with insightful modelling and analysis supporting their findings.

[Re] We appreciate that the referee describes our work as interesting, insightful and well presented. We thank this referee for providing valuable comments.

Some comments and questions:

1). Is there a particular reason that In was chosen over another system? Other seed solutions could also reduce out metal nanoparticles following electron beam irradiation.

[Re] We take In as our research system mainly for the following reasons

1. Previous in-situ liquid cell TEM study mainly focused on the nucleation and growth of chemically stable metals (e.g., Au, Pt) using their seed solutions⁸. However, the in-situ studies of metals with broad applicability in the fourth and fifth period elements are rarely reported. Representative Fe, Co, Ni seed solutions have been reported for in-situ growth of nanoparticles, but the oxide particles are normally obtained rather than the metals^{9,10}. So,

what we observed was the interface between oxide and aqueous solution rather than the interface between metal and aqueous solution. In-situ growth of In nanocrystals using InCl_3 seeds solution enables direct observation of metal-solution interfaces.

2. In is a widely used metal in the semiconductor industry, but the growth and nucleation of In nanocrystals have not been reported. An understanding of the nucleation and growth of In nanocrystal will help us to achieve the controllable synthesis of In nanocrystals.

2). Line 78,79, page 4 suggests that the contrast features indicated in Fig 1e are small In clusters, per the reconstructed FFT. Currently this seems an overly confident analysis, at least based exclusively on the single reconstructed image. Random darker contrast features can be seen along the right side of the IFFT as well, for instance. Could these not be a thickness effect? Would we expect to see signals from such small clusters of In over the contributions from the thickness of the water layer and windows? Their analysis here is supported somewhat by their later spectroscopy, however the current ordering of results suggests that the image analysis on its own is sufficient at identifying the interphase as containing In clusters.

[Re] Thanks for the comment. As a response to this comment, we have done additional experiments using an aberration-corrected TEM to image the small In clusters within the quasi-liquid phase. We added a high-resolution video (Movie S3) in the revised manuscript. With the aberration-corrected TEM and thin liquid cell, we are able to observe details of the quasi-liquid phase with super high resolution. Some representative images are shown in Supplementary Fig. 3, and some small nanoclusters are indicated by the white arrows. The quasi-liquid interphase has a higher contrast compared to the solution background, and the structural differences between the solution and quasi-liquid phase can be distinguished with no ambiguity.

Supplementary Fig. 3. Representative high-resolution TEM images of In nanocrystals with quasi-liquid phase. Small In nanoclusters/nanocrystals are indicated by white arrows.

3) For Fig 1f, some more detail on the EELS acquisition would be useful. Is there an image of the region from where it was acquired? How was motion of the nanoparticle mitigated against during spectra acquisition?

[Re] The spectra of EELS in the quasi-liquid phase and surrounding solution were taken from points A and B in Supplementary Fig. 4, respectively. We select nanoparticles that are attached to the viewing window for EELS spectrum collection. Compared to nanoparticles in solution, the nanoparticles attaching to the window are more easily detected with high contrast. Since the particles are attached to the viewing window, we do not observe particle motion during the spectrum acquisition.

Supplementary Fig. 4 HAADF-STEM image of In nanocrystal with quasi-liquid phase for EELS. EELS spectrum of quasi-liquid phase and solution are collected from points A and B, respectively.

4). There is an interesting discussion on page 5 to 6 regarding the role of the electron beam on the interphase formation, and the absence of the interphase on ex-situ prepared oxide core-shell nanoparticles. Would it be possible to introduce metallic In nanoparticles ex-situ, similarly to how they have already done with their oxides, rather than forming them in-situ by the reduction of the seed solution? This would allow the authors to distinguish between the interphase forming due to some interesting seed growth behavior in the solution, versus it being a consequence of the surface being metal or oxide.

[Re] This is a great question. We have done a series of control experiments as summarized as follows, which can address this question very well. First, we prepared In nanocrystals by wet chemical or electrochemical methods without introducing ligands and under an inert atmosphere. We found it is unavoidable that the In nanocrystals were oxidized during the sample preparation, such as during the centrifugal washing processes.

To introduce metallic In nanocrystals into liquid cell TEM, In nanocrystals synthesized through thermal evaporation were quickly transferred to the glove box with inert-gas protection. When we directly observe the surface of these In nanocrystals, we can find the quasi-liquid phase around the large In nanocrystals, as shown in Fig. R1. It is also important to note that that the interphase doesn't completely wrap these nanocrystals, which should be related to the partial oxidation of the surface In. And the interphase was not observed on the smaller nanocrystals because they could be fully oxidized quickly¹¹.

Fig. R1. In-situ observation of In nanocrystals in water. The In nanocrystals were synthesized through thermal evaporation and they were rapidly transferred into a liquid cell under inert-gas protection.

Reviewer #3:

I should note at the outset that I have little expertise in the experimental techniques employed in this research, but, that said, I find it impressive that atomistic details of a solid-liquid interface can be seen in real time.

Possibly because of the word limit and because the authors naturally want to report all the necessary checks they made, it was not always easy to get the main points of the story. Just getting a clear account would be my main recommendation. That said, here are some queries which I would encourage the authors to consider.

1) The three stages mentioned are a) the formation of an In cluster, b) the formation of quasi-liquid phase and the etching away of the core, and c) a steady state in which both the interfacial properties and the projected interfacial area of the core remained roughly constant. I am still not clear about this final stage, however. Earlier the authors stated that the interface retained its thickness up to a time of two weeks, so at this point I assumed the system was at equilibrium. Figure 2d shows the diameter of the core continually decreasing up to 400 s, so clearly the system is not in equilibrium up to that time. Can the authors clarify what the equilibrium state is? Or is equilibrium a state in which all the clusters have been etched away?

[Re] The equilibrium state is that the diameter of the metal core and the thickness of the quasi-liquid phase remain constant in absence of the electron beam or at a low electron beam current density. In the final stage, the system reaches an equilibrium, and both the interphase and the core remain stable. The particle in the final stage could keep stable because we use a low electron beam intensity, which has low influence on the solution species.

Fig. 2b, d shows the non-equilibrium process of In nanoparticles under a much higher electron beam dose. With the electron beam irradiation, many reactive species are generated in solution from radiolysis of water, and these highly reactive species can easily react with the nanoparticles^{2,12}. Therefore, we consider that the differences between the stated final stage and the results in Fig. 2b, d are due to the environment in which the particles are located has been changed because of the electron beam irradiation effects.

2) I am also not clear what chemical reaction is thought to be going on. The authors argue that hydroxide is present in the interfacial region. Is the In forming some kind of cation in order to go into solution? Some clarity here would certainly help me.

[Re] We can simply treat the reaction as $2\text{In} + \text{In}^{3+} \leftrightarrow 3\text{In}^+$, but the reaction can be more complicated in solution because of complexation effects between ions. In fact, the synthesis of indium subhalides by the reduction of InCl_3 with indium metal in various stoichiometries has been reported, and indium subhalides have been confirmed to exist as mixed-valent salt. Many mixed valence indium halides have been characterized as ionic species, including $[\text{In}^+]_6[\text{InCl}_9]^{6-}$, $[\text{In}^+]_3[\text{In}_2\text{Cl}_9]^{3-}$ and $[\text{In}^+]_3[\text{InCl}_6]^{3-}$.¹³ Therefore, the interfacial phase may contain different In ionic forms. In addition to the coordination of chloride ions with indium ions, H_2O can also be decomposed into hydroxyl radicals or OH^\cdot ,² which can enter the interphase by interacting with In or In^{3+} .

3) The authors discuss the liquid like material showing dynamical changes between amorphous and nano-crystalline. I did stare at the videos but found it hard to see where/when this was happening. The figures in the manuscript did not make this obvious, at least not to me. This is an intriguing observation but does need some backing up with evidence. It would also be nice to know what species were involved, something on the timescales of the process, etc.

[Re] Thanks for pointing this out. What we mainly discuss here is the transition between the amorphous interphase and In nanoclusters, rather than the quasi-liquid phase itself transitioning between amorphous and crystalline. The transition between amorphous interphase and In nanoclusters is shown in Fig. 3a (also see Movie S8). Through high-resolution TEM images and videos, we can find that the liquid-like phase is amorphous. It can be seen from the high-resolution video that nanoclusters can grow directly from the quasi-liquid phase. The newly formed nanoclusters have higher contrast than the quasi-liquid phase, so we refer that the small nanoclusters are small In nanocrystals. What happens during this process is a transformation from

amorphous quasi-liquid phase to crystalline In. In addition, we can see that a small nanoparticle enters the interphase and transforms into a part of the quasi-liquid phase. In this process, small In nanocrystal becomes amorphous quasi-liquid phase. In the transition between the quasi-liquid phase and the In nanoclusters can be understood based on the following equation: $2\text{In} + \text{In}^{3+} \leftrightarrow 3\text{In}^+$. When the equilibrium of the reaction is towards the left, we observe the precipitation of particles, and when the equilibrium of the reaction is towards the right, we observe the transition of the particles to the interfacial phase. We have made modifications of the main text and highlighted the corresponding descriptions to make the above points more clear.

4) The computer simulation was interesting but some more details would be helpful. Firstly how did the authors verify the methodology (e.g. how good is the functional, how confident are they that they are working at a reasonable pressure, etc)? Secondly the simulation videos had lines appearing and disappearing and some sort of colour scheme. these should be explained.

[Re] We thank the reviewer for the comments. To elucidate the formation mechanism of quasi-liquid phase, we performed the molecular dynamics calculations based on the density functional theory^{14,15}, which is well-suited for the purpose of accounting for the dynamic evolution of the quasi-liquid phase. With this method, we can obtain the movement trajectories of atoms, especially the details that cannot be uncovered in the experiments, the accurate bonding properties and the electronic properties of the atomic dynamics. In this work, ab initio molecular dynamics (AIMD) simulations were carried out in the canonical ensemble condition, i.e., using the Nose-Hoover thermostat namely^{16,17}, the constant number of particles, volume, and temperature (NVT) utilizing the periodic simple cubic simulation cells with lattice parameters set to reproduce the experimental density of the solution in the ambient.

The appearance and disappearance of lines in AIMD simulation videos is due to the set threshold length of In-In bonds which refers to the realistic In-In bonding distance. With the structure's changes, the bond length of In-In varies, which leads to the occurrence of a bond when the In-In distance is smaller than the threshold. While it disappears when the In-In distance is larger than the threshold. Regarding the colors, it is because we mark the prototypical In (the neutral system: 250, 254, 257, 262, and 268; the charged system: 210, 214, 217, 222, and 228) to facilitate the observation during the changes.

5) More explanation should be given to the charged simulation. The starting configuration would appear to be In and OH^+ ions, from the text, though I am not sure that is what was in the video. It

does seem odd to have OH^+ , though. But even so, I presume the system eventually settles down but in a way that does not resemble the experimental system, which is charge neutral, containing chloride ions. On the face of it, this is a very artificial set-up and I am not clear what conclusions can be drawn from it.

[Re] Thanks for raising this important question. First of all, the utilization of periodic boundary conditions is adopted here, and both the charged and uncharged systems are neutral. For the “charged” system, we remove 40 H from the In/ H_2O neutral system, so the starting configuration contains 40 In atoms, 40 OH and 40 H_2O molecules. Due to the spontaneous charge transfer between OH and In, the unstable OH molecules tend to get electrons from In atoms to form stable hydroxide ions on the charged simulation. This will induce a positive charge on In. It can be seen from the statistics of charges of the charged systems, the average charge of In is calculated as $+0.42 e^-$. In fact, the In charged system is approximately composed of In-ions, hydroxide ions and H_2O after settling down, which is consistent with our experimental results.

6) Just a minor query on the uncharged simulation, is the total charge on the In atoms approximately zero or is there net charge transfer to the water? Do you get hydroxide forming, for example?

[Re] The total charge on the 40 In atoms is approximately zero ($-0.0556 e^-$) on the uncharged simulation. There is no charge transfer to the water, and no hydroxide will be generated in the uncharged system.

And again, how useful is this simulation, given the absence of chloride?

[Re] First, we demonstrated that the quasi-liquid phase can also be formed in the absence of chloride in the controlled experiment, so chlorine was not considered in the theoretical calculation. We understand the interphase is a new metal-rich phase. However, we have no idea that is the quasi-liquid phase full of In-atoms, how the In, H_2O and hydroxide species interact with each other, and why the quasi-liquid phase could keep stable? All these questions are very difficult to be solved using experimental methods. From the DTF-MD, we can understand that the quasi-liquid phase is mainly made of In-ions not In atoms. Based on our calculation, we speculated that the positive charges make it difficult for In atoms to gather together, thereby making the quasi-liquid phase relatively stable.

I apologies if the answers to many of these questions are obvious, but I think the authors need to make everything a lot clearer before publication, even though I think the experimental observations

are, to my mind, impressive.

[Re] We thank this referee for the valuable comments and questions, which have made our manuscript, especially the part on theoretical calculation, more readable. The clarity of our revised manuscript has been much improved.

References

- 1 Hill, M. & Smith, F. Calculation of initial and primary yields in the radiolysis of water. *Radiat. Phys. Chem.* **43**, 265-280 (1994).
- 2 Schneider, N. M. *et al.* Electron–Water Interactions and Implications for Liquid Cell Electron Microscopy. *The Journal of Physical Chemistry C* **118**, 22373-22382, doi:10.1021/jp507400n (2014).
- 3 Ghormley, J. Lifetime of intermediates in water subjected to electron irradiation. *Radiation Research* **5**, 247-251 (1956).
- 4 Alfano, J. C., Walhout, P., Kimura, Y. & Barbara, P. F. Ultrafast transient-absorption spectroscopy of the aqueous solvated electron. *The Journal of chemical physics* **98**, 5996-5998 (1993).
- 5 Phaniendra, A., Jestadi, D. B. & Periyasamy, L. Free radicals: properties, sources, targets, and their implication in various diseases. *Indian journal of clinical biochemistry* **30**, 11-26 (2015).
- 6 Schneider, N. M. Electron beam effects in liquid cell TEM and STEM. *liquid cell electron microscopy* **140** (2016).
- 7 Loh, N. D. *et al.* Multistep nucleation of nanocrystals in aqueous solution. *Nat. Chem.* **9**, 77-82, doi:10.1038/nchem.2618 (2017).
- 8 Liao, H.-G. & Zheng, H. Liquid cell transmission electron microscopy. *Annu. Rev. Phys. Chem.* **67**, 719-747 (2016).
- 9 Liang, W.-I. *et al.* In Situ Study of Spinel Ferrite Nanocrystal Growth Using Liquid Cell Transmission Electron Microscopy. *Chem. Mater.* **27**, 8146-8152 (2015).
- 10 Yang, J. *et al.* Formation of two-dimensional transition metal oxide nanosheets with nanoparticles as intermediates. *Nat. Mater.* **18**, 970-976, doi:10.1038/s41563-019-0415-3 (2019).
- 11 Sutter, E. & Sutter, P. Size-Dependent Room Temperature Oxidation of In Nanoparticles. *The Journal of Physical Chemistry C* **116**, 20574-20578, doi:10.1021/jp305806v (2012).
- 12 Jiang, Y. *et al.* In situ study of oxidative etching of palladium nanocrystals by liquid cell electron microscopy. *Nano Lett.* **14**, 3761-3765, doi:10.1021/nl500670q (2014).
- 13 Downs, A. J. *Chemistry of aluminium, gallium, indium and thallium.* (Springer Science & Business Media, 1993).
- 14 Car, R. & Parrinello, M. Unified approach for molecular dynamics and density-functional theory. *Phys. Rev. Lett.* **55**, 2471 (1985).
- 15 Marx, D. & Hutter, J. *Ab initio molecular dynamics: basic theory and advanced methods.* (Cambridge University Press, 2009).
- 16 Nosé, S. A unified formulation of the constant temperature molecular dynamics methods. *The Journal of chemical physics* **81**, 511-519 (1984).
- 17 Hoover, W. G. Canonical dynamics: Equilibrium phase-space distributions. *Phys. Rev. A* **31**, 1695 (1985).

REVIEWERS' COMMENTS

Reviewer #1 (Remarks to the Author):

The authors have addressed all my comments and I now suggest its publication on Nature Communications.

Reviewer #2 (Remarks to the Author):

The authors have addressed all my questions. The new images included in Fig S3 look excellent.

Reviewer #3 (Remarks to the Author):

As in my previous review, despite my lack of expertise in the experimental technique, I think these are impressive results. My comments were largely based around the molecular dynamics simulations and trying to get a clearer understanding of what was going on. I would like to thank the authors for responding to my questions but I suspect we have crossed wires. I do not want to delay publication, for it is the experimental work that is the really important part of this paper, but I still think the authors need to give some more simulation explanations/motivation.

1) What is the evidence that their functional gives a good description of the system? The answer does not seem to address this but instead discusses simulation methodology. Just a reference to a previous study on such a system would be fine.

2) Calling a simulation "charged" is very confusing, considering the system is neutral. That tripped me up previously. At the very least a sentence should be added to say, for example, that the system is neutral but the conditions are chosen such that it will form cations.

3) I still do not see why this "charged" system makes physical sense. It still seems a highly artificial starting point and some justification/motivation for its choice should be given in the text.

I recommend publication but I think the article needs a few lines of explanation/motivation for aspects of the simulation, as well as some justification for the chosen density functional.

A J Masters

Response to reviewers' comments

Reviewer #1 (Remarks to the Author):

The authors have addressed all my comments and I now suggest its publication on Nature Communications.

[Re] We thank the referee's positive comments.

Reviewer #2 (Remarks to the Author):

The authors have addressed all my questions. The new images included in Fig S3 look excellent.

[Re] Thank you very much for your appreciation.

Reviewer #3 (Remarks to the Author):

As in my previous review, despite my lack of expertise in the experimental technique, I think these are impressive results. My comments were largely based around the molecular dynamics simulations and trying to get a clearer understanding of what was going on. I would like to thank the authors for responding to my questions but I suspect we have crossed wires. I do not want to delay publication, for it is the experimental work that is the really important part of this paper, but I still think the authors need to give some more simulation explanations/motivation.

[Re] Thanks for the valuable comments and suggestions.

1) What is the evidence that their functional gives a good description of the system? The answer does not seem to address this but instead discusses simulation methodology. Just a reference to a previous study on such a system would be fine.

[Re] Many metal-water interfaces and metal-ion solutions systems have been explored using density functional theory (DFT) based ab initio molecular dynamics simulations (AIMD). For the DFT calculations, the Perdew–Burke–Ernzerhof functional¹ with Grimme's dispersion correction² describes metal-water systems and ionic systems fairly well. For example, our collaborators J. Cheng et. al report the AIMD of electrified Pt (111)/water interface, unraveling the molecular-level picture and electronic structure of electric double layer for understanding the capacitive behavior of the interface water.³ In addition, the AIMD has been performed to understand the electrified interfacial water structures at atomically flat surfaces. Structural transitions of interfacial water at electrified Au single crystalline electrode surface were found, which is in agreement with the results from in-situ Raman spectroscopy.⁴

2) Calling a simulation "charged" is very confusing, considering the system is neutral. That tripped me up previously. At the very least a sentence should be added to say, for example, that the system

is neutral but the conditions are chosen such that In will form cations.

[Re] We agree with this good suggestion. We have reorganized the theoretical part to avoid calling the simulation "charged" system. In addition, we also state that all the simulation models are neutral in the revised manuscript.

3) I still do not see why this "charged" system makes physical sense. It still seems a highly artificial starting point and some justification/motivation for its choice should be given in the text.

[Re] The surface charge at the solid-liquid interface is very common for colloidal particles. The colloid and the ionic environment produce an electric net potential on the colloid surface and ionic cloud through the adsorption of ions.^{5,6} Here, we consider the In nanoparticle as a colloidal nanocrystal, and the quasi-liquid phase is similar to the ionic cloud on the particles' surface.

I recommend publication but I think the article needs a few lines of explanation/motivation for aspects of the simulation, as well as some justification for the chosen density functional.

[Re] Thanks for the valuable comments and suggestions.

References

- 1 Perdew, J. P., Burke, K. & Ernzerhof, M. Generalized gradient approximation made simple. *Phys. Rev. Lett.* **77**, 3865 (1996).
- 2 Grimme, S., Antony, J., Ehrlich, S. & Krieg, H. A consistent and accurate ab initio parametrization of density functional dispersion correction (DFT-D) for the 94 elements H-Pu. *The Journal of chemical physics* **132**, 154104 (2010).
- 3 Le, J.-B., Fan, Q.-Y., Li, J.-Q. & Cheng, J. Molecular origin of negative component of Helmholtz capacitance at electrified Pt (111)/water interface. *Science advances* **6**, eabb1219 (2020).
- 4 Li, C.-Y. *et al.* In situ probing electrified interfacial water structures at atomically flat surfaces. *Nat. Mater.* **18**, 697-701 (2019).
- 5 Hunter, R. J. *Zeta potential in colloid science: principles and applications*. Vol. 2 (Academic press, 2013).
- 6 Pfeiffer, C. *et al.* Interaction of colloidal nanoparticles with their local environment: the (ionic) nanoenvironment around nanoparticles is different from bulk and determines the physico-chemical properties of the nanoparticles. *J. R. Soc. Interface* **11**, 20130931 (2014).